# Dietary Lysine Supplementation Above Requirement Improves Carcass Traits and Enhances Pork Flavor Profiles in Finishing Pigs Under Commercial Conditions

**DOI:** 10.3390/foods14183262

**Published:** 2025-09-20

**Authors:** Jialong Liao, Fengyi Song, Boyang Wan, Haijun Sun, Jingdong Yin, Xin Zhang

**Affiliations:** 1State Key Laboratory of Animal Nutrition and Feeding, College of Animal Science and Technology, China Agricultural University, Beijing 100193, China; s20233040817@cau.edu.cn (J.L.); fysong289351@163.com (F.S.); b20223040354@cau.edu.cn (B.W.); sunhaijun411@163.com (H.S.); yinjd@cau.edu.cn (J.Y.); 2Frontiers Science Center for Molecular Design Breeding (MOE), Beijing 100193, China

**Keywords:** carcass traits, dietary lysine levels, free amino acids, pork quality, volatile flavor compounds

## Abstract

To explore the effects of dietary lysine level (DLLs) on growth performance, carcass traits, meat quality and flavor characteristics in finishing pigs under large-scale commercial farming conditions, approximately 450 Duroc × Landrace × Yorkshire crossbred finishing pigs (initial body weight: 103.65 ± 4.28 kg) were randomly assigned to four treatment groups in this study. Each group consisted of four replicate pens, with 25~30 pigs per pen. The Lys100 group received a diet formulated according to the NRC (2012) standard. The standardized ileal digestible lysine (SID Lys) levels in the diets for the Lys115, Lys130 and Lys145 groups were set at 115%, 130% and 145% of the level in the Lys100 group, respectively. The trial lasted for 31 days. The results showed that increasing DLLs by 15%, 30% or 45%, while safeguarding the lysine requirement and maintaining the ideal ratios of other essential amino acids to lysine, had no negative impact on growth performance or meat quality. For carcass traits, increasing lysine levels in diets linearly increased loin eye area (*p* = 0.018) and tended to reduce backfat thickness at the 10th rib (*p* = 0.096). Methionine and glycine contents in the longissimus thoracis (LT) muscle linearly increased with an increase in DLLs (*p* = 0.014 and 0.073, respectively). Furthermore, increasing lysine levels by 45% significantly increased the percentage of volatile flavor compounds (VOCs) belonging to nitrogen compounds (*p* = 0.040), ethers (*p* = 0.026) and aldehydes (*p* = 0.040), as well as increased contents of key VOCs, such as (E)-2-Nonenal (*p* = 0.005), (E)-2-Octenal (*p* = 0.005) and 1-Octen-3-one (*p* = 0.008), contributing to enhanced sweet, fruity, fatty and waxy flavor profiles. According to various indexes, better carcass traits and pork flavor could be achieved by increasing lysine levels by 45% in diets based on the recommended value for finishing pigs.

## 1. Introduction

In 2024, the global production of pork reached 116.02 billion tons. China, the European Union and the United States accounted for 48.91%, 18.32% and 10.93% (United States Department of Agriculture, USDA). Pork quality directly influences consumer purchasing preferences and serves as a core indicator for evaluating the profitability of pig farming [1]. Consumers increasingly demand diversity in pork flavor and a natural taste profile. However, the accumulation of flavor-related compounds in muscle, including essential amino acids and polyunsaturated fatty acids, is being neglected due to the excessive pursuit of growth rate and lean meat percentage. This leads to the homogenization of pork flavor. For instance, a reduction in lipid-soluble flavor compounds (such as aldehydes) weakens the fatty aroma and meatiness of pork [2]. Therefore, it is of great importance to find effective methods that balance growth performance with pork flavor.

Pork flavor originates from three main sources: lipid oxidation, the Maillard reaction and thiamine degradation [3]. The types and content of volatile flavor compounds (VOCs) directly determine the sensory quality of pork. Factors such as breed, nutrition, feeding methods and processing influence pork flavor. Compared to Yorkshire pigs, Laiwu pigs, a Chinese native breed, have a higher triglyceride content in their muscles, with significantly increased levels of α-ketoglutaric acid, fumaric acid and L-aspartic acid. These play an important role in enhancing the flavor of Laiwu pork [4]. The Chinese local breed Ningxiang pig has a unique sweet, fruity and floral aroma compared to Berkshire pigs, which originates from higher levels of 2-pentylfuran, pentanal, 2-(E)-octanal and acetic acid [5]. It is worth noting that dietary nutrition is the most controllable and cost-effective means for regulating pork flavor [6]. Dietary amino acid composition plays a crucial role in muscle development, fat deposition and the accumulation of metabolic by-products [7]. For instance, adding 1.5% arginine to a low-protein diet increases the accumulation of arginine in pork by 51.54%, thereby enhancing its overall acceptability [8]. Additionally, arginine exhibits synergistic effects with other amino acids. Supplementing with 1.0% L-arginine and 1.0% glutamic acid significantly increases IMF accumulation and fatty acid content, generating multiple VOCs derived from fatty acid oxidation. This enhances meat tenderness, juiciness and overall quality [9]. Concurrently, low dietary protein (14.0%) can also significantly alter the volatile compound profile of pork and increase the abundance of aldehydes, ketones and alcohols. However, these studies rarely, if ever, elucidate and analyze the specific mechanisms influencing flavor, such as the role of precursor substances in flavor formation [10].

Lysine is an essential amino acid that the body cannot produce itself and must obtain from food. It plays a crucial role in promoting protein synthesis in humans, as well as fatty acid metabolism and calcium absorption. It can even enhance immune function and reduce anxiety [11,12,13]. Lysine is also the first limiting amino acid in pig diets. Previous studies have shown that increasing the level of standard ileal digestible (SID) lysine in diets improves growth performance and linearly increases the hot carcass weight and lean meat percentage in finishing pigs [10]. Reducing the lysine/protein ratio in diets from 0.046 to 0.035 increased the marbling score of longissimus thoracis (LT) muscle and elevated intramuscular fat (IMF) content by 38.71% [14]. In finishing pigs weighing 71–123 kg, reducing the dietary ratio of SID lysine to digestible energy from 0.6 to 0.4 increased IMF content by 17.6% and improved tenderness [15]. Lysine has a sweet taste, and the carnosine formed from lysine could mask bitterness [16]. Additionally, lysine promotes carnitine synthesis, enhances fatty acid β-oxidation and improves ATP production efficiency, thereby increasing inosine monophosphate (IMP) accumulation [17], which is a key source of umami flavor in meat. Therefore, dietary lysine level (DLLs) may influence the development of pork flavor. However, studies regarding the effects of DLLs on pork flavor characteristics are currently limited. Furthermore, most studies have used small numbers of pigs, and the impact of DLLs on pork quality and flavor under large-scale commercial farming conditions requires systematic investigation.

Therefore, the objective of this study was to investigate the effects of DLLs on carcass traits, meat quality and VOC composition in pork while meeting the lysine requirements, using 450 finishing pigs. We hypothesized that increasing DLLs would alter pork flavor without negatively affecting growth performance, carcass traits or meat quality under large-scale farming conditions.

## 2. Materials and Methods

### 2.1. Experimental Design and Sample Collection

This study obtained the approval of the Institutional Animal Care and Use Committee of China Agricultural University (approval number: AW01202313-1).

Animal experiments were carried out in a commercial pig farm (Fuzhiyuan Group, Guiyang, Guizhou, China) and animals were fed ad libitum. A total of approximately 450 Duroc × Landrace × Yorkshire crossbred finishing pigs (103.65 ± 4.28 kg) were randomly divided into four treatment groups, taking into account their initial body weight (BW). There were four replicate pens of 25~30 pigs per replicate in each group. Therefore, a total of approximately 110 pigs were involved in each group. All diets were formulated according to the NRC (2012) [18]. These groups were named the Lys100, Lys115, Lys130 and Lys145 groups. In brief, in the Lys100 group, SID Lys in the pigs’ diets was at the recommended value. In the Lys115, Lys130 and Lys145 groups, SID Lys in the pigs’ diets was 115%, 130% and 145% of that in the Lys100 group, respectively. The levels of other essential amino acids were determined based on the ideal amino acid ratio (NRC, 2012). Dietary composition and nutrient levels are listed in Table 1. The finishing pigs had ad libitum access to feed and water during the experimental period of 31 days. The management and immunization procedures were carried out according to the company’s regulations. Feed intake was recorded daily. All pigs were weighed individually at the beginning and end of the experiment. Initial and terminal BW were recorded. The average daily gain (ADG) and ratio of feed to gain were calculated and analyzed for each pen.




A
D
G
k
g
/
d
=
[
F
i
n
a
l

B
W
(
k
g
)
−
I
n
i
t
i
a
l

B
W
k
g
]
/
31
d
A
v
e
r
a
g
e

d
a
i
l
y

f
e
e
d

i
n
t
a
k
e
k
g
/
d
(
A
D
F
I
)
=
T
o
t
a
l

f
e
e
d

i
n
t
a
k
e
k
g
/
31
d
R
a
t
i
o

o
f

f
e
e
d

t
o

g
a
i
n
=
A
D
F
I
k
g
/
d
/
A
D
G
k
g
/
d
.



After the feeding period, one barrow and one gilt of the average BW per pen were picked up (*n* = 8, female to male ratio = 1:1). The slaughter weight was approximately 136 kg. After fasting for 12 h with free access to water, the pigs were transported to a designated slaughterhouse near the farm. After resting for at least four hours in holding pens with shower facilities, the pigs were humanely stunned using a high-frequency electrical system. A throat puncture was performed immediately while the animals were unconscious to allow for bleeding. To access meat quality, samples of approximately 100 g were obtained from the left LT of the carcass approximately 30 min after slaughter, located between the 10th and 12th ribs and maintained at 4 °C. Another piece of LT muscle (~100 g) was subjected to chemical composition analysis.

### 2.2. Carcass Traits

After slaughter, the carcass traits of pigs were measured (*n* = 8). The weight of each carcass was measured after exsanguination, evisceration and removal of the head and hooves. Dressing percentage was calculated as follows: Dressing percentage (%) = carcass weight/live weight × 100, where live weight refers to the BW of pigs after 12 h of fasting with free access to water, before slaughter. Backfat thickness was measured at five positions using a vernier caliper with a precision of 0.01 mm: the thickest part of the shoulder, the 6th–7th rib interface, the 10th rib, the last rib and the last lumbar vertebra. The average backfat thickness was calculated using three of these measurements: shoulder backfat depth, backfat depth at the last rib and lumbosacral backfat depth. At the 10th rib, the height and width of the LT muscle (loin eye) were measured using a vernier caliper. The loin eye area was calculated using the formula following:
$$Loin\;eye\;area\left({cm}^{2}\right)=height\left(cm\right)\times width\left(cm\right)\times 0.7.$$

### 2.3. Meat Quality Assessment

The meat quality of fresh LT was analyzed at 45 min and 24 h post-slaughter (*n* = 8). At 24 h post-slaughter, meat color and marbling were scored according to the National Pork Producer Council criteria: a score of 6.0 denotes a deep purplish-red color, while a score of 1.0 indicates an overly pale white shade. An SPK pH meter (model pH-star, DK2730, Herlev, Denmark) was used to assess pH at 4 °C, with readings taken at both 45 min and 24 h post-slaughter. The pH meter was calibrated beforehand using pre-cooled standard buffers (pH 4.01 and 7.00) at 4 °C, following standardized protocols. After undergoing a 30 min blooming process, the meat color was evaluated using a Konica Minolta CR-410 colorimeter (Konica Minolta, Osaka, Japan) equipped with a 50 mm measuring aperture and illuminated by a standardized xenon lamp. The instrument was set up with a D65 light source and a 2° standard observer to determine lightness (L*), redness (a*) and yellowness (b*) values according to the CIE Lab system. The chroma (c*) and hue angles (h°) were calculated as follows:
$c*=\sqrt{({a*)}^{2}+{(b*)}^{2}}$,
 h°=arctan(b*/a*) [19,20]. Each piece of meat was measured three times, and the average value was taken. Prior to use, the device was calibrated against a white reference tile.

Drip loss was determined by initially weighing fresh LT samples, sealing them in airtight lock bags and storing them at 4 °C for 24 h. After this period, the samples were weighed again to calculate the drip loss percentage [21]. To determine cooking loss, pre-weighed muscle samples were steamed in a 70 °C water bath for 30 min in one batch. After cooling to room temperature, the samples were reweighed to calculate the cooking loss percentage [22]. After calculation, the same piece of meat was cut into 10 cubes for shear force measurement. Shear force was tested on cylindrical cores (1.27 cm diameter) taken from the cooked samples using a C-LM3B digital muscle tenderness meter (Tenovo, Beijing Tianxiang Feiyu Technology Co., Beijing, China), which was fitted with a strain-gauge pressure sensor and operated at 300 mm/min [23].

### 2.4. Chemical Composition of Meat

Approximately 10 g of LT muscle samples (*n* = 8) were accurately weighed. Moisture content was determined by vacuum freeze-drying the samples at −50 °C under 0.1 mbar for 48 h (Christ Alpha 1–4 LDplus, Martin Christ Gefriertrocknungsanlagen GmbH, Osterode am Harz, Germany), followed by reweighing. Moisture content was calculated as follows:
$\left[\left(Initial\;weight-Freeze\;dried\;weight\right)/Initial\;weight\right]\times 100\%$. For crude protein (CP) determination, freeze-dried muscle samples were ground into a homogeneous powder and then analyzed via the Kjeldahl method [24] using a Kjeltec 8400 Auto Analyzer (FOSS Analytical A/S, Hillerød, Denmark). CP content was calculated by multiplying the nitrogen content by a conversion factor of 6.25. IMF was extracted using a Budwi Extraction System B-11 (Budwi, Lausanne, Switzerland) based on the Soxhlet extraction principle [25]. Ground freeze-dried samples were wrapped in filter paper thimbles extracted with petroleum ether (boiling range 30–60 °C) for 6 h, then dried at 105 °C for 1 h and cooled in a desiccator before weighing. IMF content was expressed as a percentage relative to the initial fresh weight of the muscle.

### 2.5. Analysis of Amino Acid Composition

The amino acid composition of the LT muscle was measured using the AOAC (2007) standard method [26] (*n* = 8). First, approximately 0.3 g of freeze-dried muscle was weighed and pulverized. This was then mixed with 5 mL of 20% methanol and 8 μL of internal standard (D-phenylalanine) by vortexing. This mixture was sonicated at room temperature for 5 min, left to stand at room temperature for 1 min and then vortexed; this cycle was repeated six times until the total sonication time reached 30 min. After standing for 2 h, the mixture was centrifuged at 13,000 rpm for 10 min at 4 °C. Subsequently, 400 μL of the supernatant was transferred to a 1.5 mL centrifuge tube and vacuum-dried at 45 °C for 3 h using a rotary evaporator. Finally, 100 μL of borate buffer solution and 20 μL of AccQ Tag derivatizing reagent (Kairos, **Los Angeles,**USA) were added. Analysis of free amino acids in muscle samples was conducted via ion-exchange chromatography using an Amino Acid Analyzer L-8900 (Hitachi, Tokyo, Japan). Tryptophan was quantified separately via high-performance liquid chromatography (HPLC 1200 Series, Agilent, Santa Clara, CA, USA) using alkaline hydrolysis.

### 2.6. Determination of Volatile Compounds

VOCs were extracted using solid-phase microextraction (SPME) (*n* = 7–8) [27]. A 500 mg sample of LT muscle was placed into a headspace vial along with 10 μL of 1 mg/L deuterated n-hexanol-d13 (internal standard). The sample was then incubated at 80 °C for 10 min. The SPME fiber was conditioned at 270 °C for 10 min before being transferred to the sample vial. Adsorption proceeded for 25 min at 80 °C. The loaded SPME fiber was then transferred to the gas chromatography injection port and desorbed at 250 °C for 5 min. The conditioning step was then repeated. Finally, 10 μL of n-alkanes were added to the vial to complete the extraction process.

Subsequent analysis was performed using comprehensive two-dimensional gas chromatography coupled with time-of-flight mass spectrometry (GC × GC-TOFMS). Separation was achieved using a gas chromatograph from Agilent Technologies (Palo Alto, CA, USA) equipped with primary and secondary columns. High-purity helium carrier gas flowed at a rate of 1.0 mL/min. The temperature program was as follows: the primary column was held at 50 °C for 2 min, then ramped to 230 °C at 5 °C/min and held for a further 5 min. The secondary column and modulator temperature programs were maintained at temperatures 5 °C and 20 °C higher than the primary column, respectively.

Mass spectra were acquired using a LECO Pegasus BT 4D mass spectrometer (LECO, St. Joseph, MI, USA) with the following parameters: transfer line temperature 250 °C, ion source temperature 250 °C, acquisition rate 200 spectra/s, electron ionization energy 70 eV and detector voltage 1960 V. Compound annotation was performed using Chroma TOF (version 5.0) software to identify compound names, retention times and peak areas. Unsupervised principal component analysis (PCA) was performed using the ropls package in R (version 4.2.0) to visualize natural clustering of samples from the Lys100 and Lys145 groups. Relative contents of VOCs were calculated using the semi-quantitative method with an internal standard. Compounds meeting the criteria of *p* < 0.05 and VIP > 1 were defined as differential VOCs. The relative odor activity value (ROAV) was calculated as follows:
$ROAV\left(A\right)=100\times \left[Relative\;Content\left(A\right)/Odor\;Threshold\left(A\right)\right]/\left[Relative\;Content\left(stan\right)/Odor\;Threshold\left(stan\right)\right]$. Where A represents the target compound and stan represents the compound with the highest relative content/odor threshold value. Sensory characteristics of all compounds were analyzed based on the FlavorDB database. A sensory flavor radar chart was generated for the top 10 sensory attributes ranked by content.

### 2.7. Statistical Analysis

Data are expressed as mean ± SEM. Unpaired two-tailed Student’s *t*-test and one-way ANOVA procedures were conducted using SAS (v.9.2, SAS Institute, Cary, NC, USA) to compare differences between two or four groups, respectively. The differences between the four groups was analyzed using a mixed linear model in SAS as follows: Yij = μ + Ti + uij + eij, where Yij is the dependent variable; μ is the overall mean; Ti is the fixed treatment effect; uij is the random effect of the pen within the dietary treatment; and eij is random error. Linear and quadratic regression analyses were performed to assess the effect of expository dose. Means were considered significantly different when *p* < 0.05 and as a trend when 0.05 ≤ *p* ≤ 0.10. Data visualization was implemented using R (version 4.0.3) and GraphPad Prism software (version 8.0).

## 3. Results

### 3.1. Growth Performance and Carcass Traits

There were no significant differences in the final BW, ADG, ADFI or ratio of feed to gain among the four groups of finishing pigs with varied DLLs (Table 2).

In terms of carcass traits, the loin eye area significantly increased linearly with lysine levels in diets (*p* = 0.018, Table 3). DLLs induced trends in altering the backfat depth at the lumbar vertebra (*p* = 0.067). The backfat depth in the 10th rib presented a trend of linear decrease with the increase in lysine levels (*p* = 0.096).

### 3.2. Meat Quality

As shown in Table 4, the dietary treatments failed to affect the pH values at 45 min and 24 h, meat color (L*, a*, b*, C*, h° values) at 24 h postmortem, drip loss, cooking loss, shear force, flesh color score, marbling score and IMF content in finishing pigs (*p* > 0.05, Table 4). CP content decreased significantly in a linear and quadratic manner with DLLs (*p* = 0.031 and 0.005). Compared to the Lys100 group, the CP content in meat was significantly decreased in the Lys130 group.

### 3.3. Free Amino Acid Content in LT Muscle

As shown in Table 5, DLLs had no effect on the content of essential amino acids, non-essential amino acids, flavor amino acids, or total amino acids in the LT muscle (*p* > 0.05). In addition, methionine content in LT muscle increased linearly across all treatments (*p* = 0.014, Table 5). Glutamic acid and glycine exhibited quadratic and linear increases, respectively (*p* = 0.058 and 0.073, respectively; Table 5).

### 3.4. Types of Volatile Flavor Compounds and an Overall Analysis of Sensory Flavor Characteristics

Compared to the Lys100 group, the Lys145 group exhibited the best carcass traits and an increase in the levels of methionine and glycine, which are important precursors of pork flavor. To further demonstrate the impact of dietary lysine on pork flavor, differences in VOCs and flavor profiles were analyzed using pork samples from the Lys100 and Lys145 groups. PCA revealed significant separation between the two groups (Figure 1A). A total of 2957 VOCs were identified (Appendix A) and categorized into 17 chemical groups (Table 6). The five categories with the highest proportions were alcohols, aldehydes, benzenoids, esters and hydrocarbons. Compared with the Lys100 group, the Lys145 group had significantly higher percentages of aldehydes (*p* = 0.040), nitrogen compounds (*p* = 0.040) and ethers (*p* = 0.026). Furthermore, an increase in DLLs tended to elevate the percentages of alcohols, benzenoids and esters in meat (*p* = 0.058, 0.061 and 0.093, respectively).

A sensory flavor radar chart was constructed using the FlavorDB database. The results showed that the overall sensory flavor characteristics of both groups were dominated by sweet, fruity, ethereal, fatty, waxy, strong, butter, alcoholic, creamy and buttery flavors. At the same time, there was little difference in overall sensory flavors between the two groups, but the sweet, fruity, waxy and fatty profiles in the Lys145 group were enhanced (Figure 1B).

### 3.5. Analysis of Differential Volatile Flavor Compounds

Following the exclusion of missing values, 584 VOCs were retained for subsequent analysis (Appendix A). Compared with the Lys100 group, the Lys145 group showed significant upregulation of 87 VOCs and downregulation of 86 VOCs (Appendix A). The upregulated VOCs were mainly hydrocarbons (19/87), followed by ketones (13/87), organoheterocyclic compounds (12/87), benzenoids (9/87) and alcohols (8/87). Notably, the downregulated VOCs were predominantly categorized into hydrocarbons and alcohols, accounting for 54.7% of the total downregulated compounds (Figure 1C). The interaction between differential VOCs and sensory flavor profiles is depicted in Figure 1D. The top 10 flavor characteristics were green, fruity, waxy, fatty, sweet, fresh, earthy, herbal, rose and rummy. Except for cedarwood terpenes and 5-ethyl-2,4-dimethylthiazole, all other VOCs in the figure were significantly upregulated in the Lys145 group, with the upregulated substances displaying multiple flavor characteristics.

The key flavor compounds and their odor characteristics in different pork samples were evaluated using ROAVs. The results showed that three up-regulated VOCs had ROAVs greater than 1, including 1-octen-3-one, (E)-2-nonenal and (E)-2-octenal. Among them, (E)-2-nonenal had the highest ROAV, exhibiting flavor characteristics such as waxy, green and fatty, followed by (E)-2-octenal, which presented flavor characteristics including fresh, green, waxy and fatty, while 1-octen-3-one showed an earthy flavor characteristic. Compared with the Lys100 group, the contents of these three VOCs were all higher in the Lys145 group (Appendix A).

## 4. Discussion

Current research mainly focuses on ensuring the growth performance of pigs through the balance of dietary amino acids [28,29,30]. However, studies on the effect of DLLs on pork flavor characteristics are limited. Recently, Pan et al. found that increasing DLLs linearly increased the content of sweet amino acids in soleus muscles of growing-finishing pigs (33–125 kg) and reduced the content of bitter amino acids, while also increasing the content of IMP. When DLLs were at 90% of the recommended value, it was beneficial to flavor formation and improved meat quality [31]. Nevertheless, there is still a lack of clear description of the flavor characteristics of pork. It should also be noted that a DLL below the recommended level may negatively affect the growth performance of pigs. In this study, it was found that increasing DLLs had no significant effect on ADG, ADFI, or the ratio of feed to gain when the lysine requirements of finishing pigs were met and the ideal ratio of other essential amino acids to lysine was maintained.

By contrast, carcass traits were more sensitive to changes in lysine levels. Loin eye area increased linearly with the increasing lysine levels. Lysine is a core driver of protein synthesis, promoting satellite cell differentiation by activating the mTORC1 signaling pathway and increasing skeletal muscle mass [32]. Furthermore, lysine has been reported to induce satellite cell migration by activating the FAK pathway [33] and act as a ligand for Fizzled, thereby activating the classical Wnt/β-catenin signaling pathway and promoting satellite cell proliferation and skeletal muscle growth [34]. Interestingly, the backfat thickness at the 10th rib decreased with increasing lysine levels. AMPK is an energy sensor whose activation promotes fatty acid oxidation and inhibits fat synthesis [35]. The expression levels of AMPKα1, SIRT1 and PGC-1α were significantly reduced under lysine restriction, leading to inhibited lipolysis [36]. Conversely, a high-lysine diet enhanced the activity of antioxidant enzymes in the intestine of grass carp by activating the mTOR signaling pathway and inhibiting the p38 MAPK signaling pathway [37]. However, in mice with obesity induced by a high-fat diet, adding lysine to the diet had no effect on BW or white adipose tissue weight [38], suggesting that the impact of lysine on fat deposition may depend on the species and type of diet. In this study, the 10th rib fat thickness in finishing pigs showed a linear decrease with increasing lysine levels. This fills the gap between basic lysine research in other animals and applied flavor optimization in swine production. It should be noted that backfat thickness showed strong decreasing trends that did not reach strict statistical significance. This may be due to the inherent variability in commercial populations and the sample size, which may have limited our ability to detect more subtle effects on all measured parameters.

Previous studies have shown that dietary lysine restriction increases IMF deposition in lean-type pigs, but also reduces feed conversion efficiency and lean meat percentage [39,40]. In this study, however, there were no significant differences in meat quality parameters, including IMF content, among the four groups. Therefore, while meeting the lysine requirements of finishing pigs, DLLs do not affect IMF content. Additionally, CP content first decreased and then increased with increasing DLLs, with the lowest value observed in the Lys130 group. This suggests that the effect of lysine levels on muscle protein synthesis may exhibit a ‘threshold effect’. Once this threshold is exceeded, protein synthesis signaling pathways are activated by a sufficient supply of substrates [41], leading to enhanced muscle protein deposition efficiency.

The formation of pork flavor is a complex process involving the metabolic conversion of flavor precursors such as amino acids and lipids into VOCs, with the final flavor being determined by the synergistic effects of multiple VOCs [42,43]. The composition of free amino acids in muscle directly influences the development of sensory flavors such as umami and sweetness [44]. In this study, the levels of methionine and glycine in muscle were highest in the Lys145 group. Methyl mercaptan and dimethyl sulfide, the metabolic intermediates of methionine, are important contributors to flavor characteristics [45]. Glycine, which contributes to sweetness, exhibited a linear increase. For finishing pigs, adding 0.57% glycine to a low-protein diet can increase muscle levels of glycine, tyrosine and glutamic acid and improve flavor scores [46]. In this study, the increase in muscle glycine content may be related to the synergistic regulation of amino acid metabolism. Lysine levels may indirectly affect glycine accumulation by influencing transamination or carbon skeleton allocation [47]. However, the direct contribution of increased methionine/glycine to flavor should also be investigated in the future.

By analyzing the VOCs in pork, we found that the proportions of nitrogen compounds, ethers and aldehydes were significantly higher in the Lys145 group than in the Lys100 group. Nitrogen compounds primarily originate from the Maillard reaction, protein degradation, and biosynthesis. These compounds have high flavor intensity and low thresholds, exhibiting flavor characteristics such as roasted and burnt aromas [48]. Ethers are commonly used as flavor enhancers, capable of intensifying sweet, fruity and floral aromas [49]. Aldehydes are products of fat oxidation or amino acid degradation with extremely low thresholds and diverse flavor characteristics. For example, short-chain saturated aldehydes and unsaturated aldehydes contribute a fresh grassy aroma; long-chain saturated aldehydes primarily exhibit fatty and waxy aromas; and aromatic aldehydes present sweet and fruity aromas [50,51]. Notably, the differential VOCs include six aldehydes, all of which are upregulated in the Lys145 group. Consistent with the changes in VOCs, the Lys145 group exhibits stronger flavor characteristics, such as sweet, fruity, waxy and fatty aroma, compared to the Lys100 group. ROAV can be used to assess the extent to which VOCs influence the overall flavor, with compounds having ROAV ≥ 1 being considered key flavor components [52]. In this study, 1-Octen-3-one, (E)-2-Nonenal and (E)-2-Octenal all had ROAVs greater than 1 and were upregulated in the Lys145 group. 1-Octen-3-one primarily exhibits mushroom and earthy flavors. Both 1-Octen-3-one and (E)-2-Octenal are key aromatic compounds in pork broth [53]. (E)-2-Octenal is typically associated with fatty, grassy and meat aromas. Compared to Berkshire pork, Ningxiang pork has a higher ROAV for (E)-2-octenal, which is considered a key aromatic marker that distinguishes Ningxiang pork from Berkshire pork [5]. It is important to note that linoleic acid is cleaved by 13-hydroperoxide to produce 1-Octen-3-one [54]. (E)-2-Nonenal has an extremely low threshold and is a specific product of oleic acid oxidation [55], while (E)-2-Octenal is generated from the oxidation of linoleic acid [56]. Therefore, all three of these compounds may be related to enhanced lipid oxidation, and oxidation markers should be measured in future research. Furthermore, the VOC analysis was limited to the Lys100 and Lys145 groups due to high cost. Including all groups would provide a more complete dose–response relationship. In this study, a 45% increase in lysine may raise feed costs, so an economic analysis is needed to balance flavor improvement with production costs. Its application requires large-scale trials to validate economic benefits. The pros and cons of increasing the lysine levels in the pigs’ diets are summarized in Appendix A. Moreover, our results are based on Duroc × Landrace × Yorkshire pigs, which are widely used in commercial production. However, responses may differ in local breeds due to genetic variations in metabolism and fat deposition.

## 5. Conclusions

In summary, while safeguarding the lysine requirement and maintaining the ideal ratios of other essential amino acids to lysine, increasing DLLs had no negative effect on growth performance and meat quality in finishing pigs. Increasing lysine levels in diets increased loin eye area and reduced backfat thickness. Higher levels of methionine and glycine content were observed in the meat from the Lys145 group. Moreover, a 45% increase in lysine levels enhanced sweet, fruity, fatty and waxy flavor profiles. This was primarily due to elevated percentages of VOCs belonging to nitrogen compounds, ethers and aldehydes, as well as increased contents of key VOCs, such as (E)-2-Nonenal, (E)-2-Octenal and 1-Octen-3-one. This study deepens our understanding of how DLLs affect carcass traits, meat quality and flavor in finishing pigs, providing valuable insights for optimizing nutritional regulation to improve pig production under commercial farming conditions.

## Figures and Tables

**Figure 1 foods-14-03262-f001:**
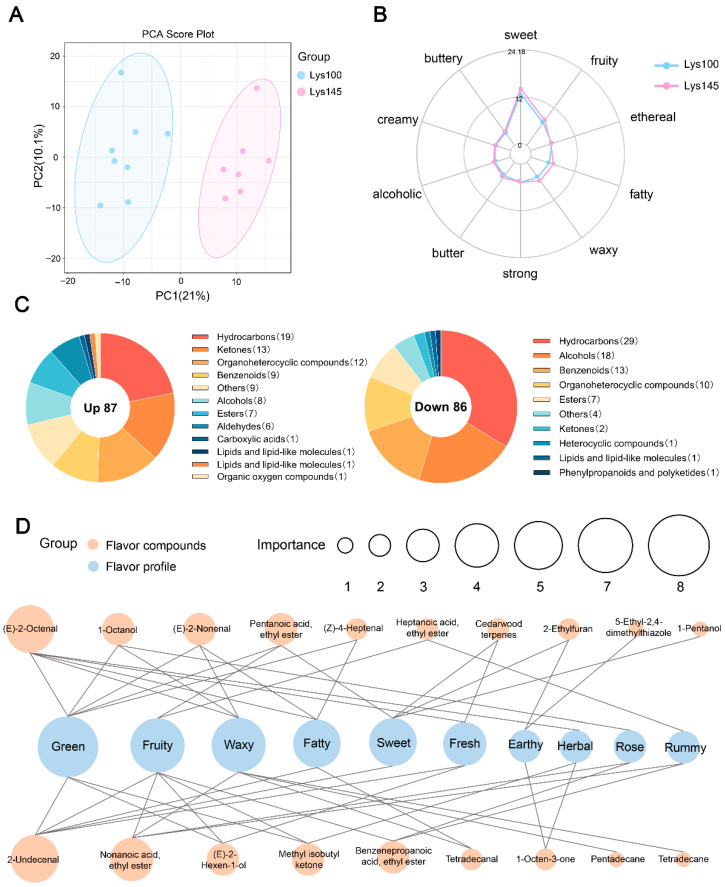
Effects of dietary lysine levels on volatile flavor compounds (VOCs) in finishing pigs of Lys100 and Lys145 groups. (**A**) Principal components analysis of all identified VOCs, *n* = 8 for Lys100 group and *n* = 7 for Lys145 group. (**B**) Sensory flavor profile depicted in a radar chart. (**C**) Classification of VOCs that are up-regulated or down-regulated in the Lys145 group relative to the Lys100 group. (**D**) Network illustrating associations between the top 10 sensory attributes and differential VOCs. Sensory attributes are shown as blue circle, with size indicating the number of connected VOCs, reflecting attribute importance. The orange circle represents VOCs and the size of the orange circle indicates the extent to which the sensory characteristics are connected to the VOCs. The top 10 sensory features were used to create the network diagram.

**Table 1 foods-14-03262-t001:** The composition and nutrient content of the experimental diets provided for finishing pigs (as fed basis, %).

Items	Lys100	Lys115	Lys130	Lys145
Ingredients				
Corn	60.41	60.20	60.04	60.04
Soybean meal	1.00	0.80	0.50	0.00
Wheat flour	15.00	15.00	15.00	15.00
Wheat bran	13.00	13.00	13.00	13.00
Rice bran	4.00	4.00	4.00	4.00
Vinasse	3.00	3.00	3.00	3.00
Soybean oil	0.40	0.40	0.40	0.40
Limestone	0.90	0.90	0.90	0.90
Dicalcium phosphate	0.15	0.15	0.15	0.15
Salt	0.30	0.30	0.30	0.30
Calcium hydrogen phosphate	0.45	0.45	0.45	0.45
Magnesium oxide	0.20	0.20	0.20	0.20
Guanidineacetic acid	0.10	0.10	0.10	0.10
L-Lysine HCl	0.60	0.80	1.00	1.20
DL-Methionine	0.00	0.03	0.08	0.14
L-Threonine	0.13	0.20	0.26	0.33
L-Tryptophan	0.01	0.03	0.05	0.08
Valine	0.00	0.05	0.12	0.20
Isoleucine	0.05	0.09	0.15	0.21
Vitamin/mineral premix ^1^	0.30	0.30	0.30	0.30
Total	100.00	100.00	100.00	100.00
Calculated values, %				
Net energy, kcal/kg	2473.00	2478.00	2484.00	2492.00
Crude protein	10.83	11.02	11.20	11.32
SID Lysine ^2^	0.61	0.70	0.80	0.88
SID Methionine	0.18	0.21	0.26	0.31
SID Methionine + cysteine	0.39	0.42	0.47	0.52
SID Threonine	0.40	0.46	0.52	0.58
SID Tryptophan	0.11	0.13	0.15	0.17
SID Valine	0.42	0.47	0.53	0.59
SID Isoleucine	0.34	0.38	0.43	0.48
Calcium	0.47	0.47	0.47	0.46
Available phosphorus	0.22	0.22	0.22	0.22

^1^ Premix provided these amounts of vitamins and minerals per kilogram on an as-fed basis for finishing pigs: vitamin A, 6450 IU; vitamin D3, 2250 IU; vitamin E, 18 IU; vitamin K3, 1.5 mg; vitamin B1, 1.65 mg; vitamin B2, 6 mg; vitamin B6, 3 mg; vitamin B12, 0.12 mg; biotin, 0.16 mg; folic acid, 1.95 mg; pantothenic acid, 13.95 mg; nicotinic, 19.5 mg; Cu (CuCl_2_), 12.51 mg; Fe (FeSO_4_·H_2_O), 105 mg; Mn (MnSO_4_·H_2_O), 41.48 mg; Zn (ZnSO_4_ ·H_2_O), 61.66 mg; I (Ca(IO_3_)_2_), 0.33 mg; Se (Na_2_SeO_3_), 0.38 mg. ^2^ Amino acids are indicated as standardized ileal digestible amino acids.

**Table 2 foods-14-03262-t002:** Effects of dietary lysine levels on growth performance of finishing pigs (*n* = 4).

Items	Treatments				SEM	*p*-Value		
	Lys100	Lys115	Lys130	Lys145		ANOVA	Linear	Quadratic
Initial weight, kg	103.96	103.96	103.58	103.08	4.282	0.995	0.883	0.853
Final weight, kg	134.79	136.54	137.27	135.54	4.024	0.973	0.873	0.678
Average daily gain, kg/d	0.99	1.05	1.09	1.05	0.036	0.292	0.216	0.155
Average daily feed intake, kg/d	3.51	3.47	3.59	3.62	0.093	0.594	0.254	0.707
Ratio of feed to gain	3.54	3.32	3.31	3.47	0.157	0.627	0.734	0.223

**Table 3 foods-14-03262-t003:** Effects of dietary lysine levels on carcass traits of finishing pigs (*n* = 8).

Items	Treatments				SEM	*p*-Value		
	Lys100	Lys115	Lys130	Lys145		ANOVA	Linear	Quadratic
Live body weight, kg	135.75	138.00	132.69	139.94	3.786	0.534	0.494	0.228
Hot carcass weight, kg	102.95	98.09	100.49	105.90	2.947	0.411	0.453	0.133
Dressing percentage, %	75.81	75.47	75.75	75.68	0.526	0.183	0.672	0.201
Back fat depth, mm								
Shoulder fat thickness	33.68	31.42	34.27	32.21	2.114	0.767	0.874	0.964
The 6th to 7th rib fat thickness	20.45	20.37	23.10	19.48	1.503	0.373	0.984	0.252
The 10th rib fat thickness	19.56	17.77	19.50	15.20	1.474	0.157	0.096	0.405
The last rib fat thickness	18.90	17.99	20.80	18.86	1.557	0.631	0.702	0.742
Lumbosacral fat thickness	8.82	7.82	9.92	7.12	0.739	0.067	0.373	0.238
Average backfat depth	20.46	19.08	21.66	19.40	1.148	0.395	0.914	0.703
Loin eye area, cm^2^	49.52	52.88	52.78	56.37	1.817	0.093	0.018	0.957

**Table 4 foods-14-03262-t004:** Effects of dietary lysine levels on meat quality of finishing pigs (*n* = 8).

Items	Treatments				SEM	*p*-Value		
	Lys100	Lys115	Lys130	Lys145		ANOVA	Linear	Quadratic
pH_45min_	5.88	5.77	5.84	5.88	0.121	0.914	0.896	0.573
pH_24h_	5.65	5.69	5.58	5.66	0.043	0.294	0.725	0.577
Meat color, 24 h								
L*_24h_	51.56	53.53	51.96	51.52	0.851	0.315	0.663	0.174
a*_24h_	15.99	15.64	16.17	15.64	0.364	0.662	0.753	0.814
b*_24h_	5.90	6.63	7.16	6.13	0.457	0.233	0.551	0.063
C*	17.06	17.02	17.73	16.83	0.420	0.472	0.924	0.276
h°	22.16	22.89	23.85	21.22	1.273	0.444	0.903	0.120
Drip loss, %	2.37	2.10	1.88	1.86	0.291	0.583	0.194	0.678
Cooking loss, %	23.11	21.58	25.05	22.86	1.493	0.447	0.692	0.831
Shear force, N	81.92	88.30	87.02	87.72	7.543	0.932	0.645	0.714
Flesh color score	4.00	3.75	3.88	3.81	0.203	0.848	0.644	0.653
Marbling score	2.81	2.94	2.50	2.69	0.244	0.624	0.903	0.242
Crude protein, %	23.59 ^a^	22.62 ^ab^	22.10 ^b^	22.85 ^ab^	0.263	0.004	0.031	0.005
Intramuscular fat, %	2.73	2.97	2.85	2.85	0.392	0.973	0.863	0.694

^a,b^ Within a row, means without a common superscript differ at *p* < 0.05.

**Table 5 foods-14-03262-t005:** Effects of dietary lysine levels on free amino acid composition in *longissimus thoracis* muscle of finishing pigs (*n* = 8), μg/g fresh meat weight.

Items	Treatments				SEM	*p*-Value		
	Lys100	Lys115	Lys130	Lys145		ANOVA	Linear	Quadratic
Lysine (Lys)	51.84	52.14	55.19	53.46	3.841	0.926	0.657	0.793
Methionine (Met)	77.84	80.35	84.82	89.04	3.141	0.088	0.014	0.792
Tryptophan (Trp)	28.07	25.75	28.28	27.85	2.583	0.892	0.872	0.723
Threonine (Thr)	23.34	22.98	26.87	24.99	1.609	0.325	0.234	0.646
Valine (Val)	12.22	14.46	14.90	14.28	0.908	0.184	0.113	0.128
Isoleucine (Ile)	15.82	14.53	14.14	15.68	1.142	0.665	0.886	0.235
Leucine (Leu)	21.63	21.64	23.56	21.13	2.096	0.854	0.971	0.576
Arginine (Arg)	18.71	19.00	18.75	21.14	1.382	0.568	0.277	0.464
Histidine (His)	26.97	27.14	31.45	28.04	2.351	0.518	0.483	0.452
Phenylalanine (Phe)	30.01	26.45	23.98	24.36	3.985	0.704	0.280	0.623
Alanine (Ala)	39.43	42.68	41.51	47.94	3.292	0.324	0.118	0.633
Asparagic acid (Asp)	34.54	34.78	45.02	45.06	9.015	0.721	0.318	0.991
Glutamic acid (Glu)	19.66	30.05	22.41	17.48	3.896	0.143	0.425	0.058
Glycine (Gly)	12.57	13.76	14.07	14.49	0.715	0.282	0.073	0.593
Proline (Pro)	15.69	16.49	16.53	16.12	1.201	0.963	0.813	0.628
Serine (Ser)	26.95	25.65	27.11	29.15	1.647	0.526	0.282	0.325
Tyrosine (Tyr)	27.72	28.64	28.71	28.82	1.304	0.933	0.577	0.761
Cystine (Cys)	0.004	0.006	0.003	0.004	0.001	0.314	0.325	0.804
Flavor amino acid ^1^	124.91	140.27	141.77	146.11	11.332	0.584	0.216	0.635
Essential amino acid ^2^	306.45	304.46	321.93	319.96	16.775	0.832	0.459	0.992
Non-essential amino acid ^3^	176.56	192.05	195.37	199.06	12.922	0.631	0.235	0.656
Total amino acid	483.01	496.51	517.30	519.02	27.311	0.755	0.306	0.832

^1^ Flavor amino acid = Glu + Asp + Ala + Arg + Gly. ^2^ Essential amino acid = Lys + Met + Trp + Thr + Val + Ile + Leu + Arg + His + Phe. ^3^ Non-essential amino acid = Ala + Asp + Glu + Gly + Pro + Ser + Tyr + Cys.

**Table 6 foods-14-03262-t006:** Percentages of volatile categories of identified volatile flavor compounds in longissimus thoracis muscle of finishing pigs (*n* = 7–8), %.

Items	Lys100	Lys145	SEM	*p*-Value
Alcohols	9.54	10.31	0.227	0.058
Aldehydes	7.13	8.01	0.188	0.040
Benzenoids	6.92	7.66	0.186	0.061
Esters	6.36	7.11	0.196	0.093
Hydrocarbons	6.34	6.61	0.173	0.191
Carboxylic acids	6.14	6.70	0.184	0.123
Organic oxygen compounds	5.92	5.38	0.165	0.734
Ketones	5.88	5.62	0.152	0.473
Lipids and lipid-like molecules	5.60	5.28	0.153	0.560
Organic acids and derivatives	5.45	5.15	0.151	0.560
Organic 1,3-dipolar compounds	5.38	5.04	0.150	0.595
Heterocyclic compounds	5.34	4.53	0.146	0.935
Organohalogen compounds	4.95	4.25	0.146	0.998
Sulfur compounds	4.76	3.63	0.131	0.492
Nitrogen compounds	4.61	5.38	0.142	0.040
Ethers	4.19	5.13	0.144	0.026
Others	5.52	4.20	0.149	0.479

## Data Availability

The original contributions presented in this study are included in the article/Appendix A. Further inquiries can be directed to the corresponding author.

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
