# Peer review of "Dietary Lysine Supplementation Above Requirement Improves Carcass Traits and Enhances Pork Flavor Profiles in Finishing Pigs Under Commercial Conditions"

_foods, 2025, doi:10.3390/foods14183262_

Round 1
Reviewer 1 Report
Comments and Suggestions for Authors
The manuscript is logically structured, with clear methods and results. The discussion ties findings to existing literature reasonably well. However, there are some methodological limitations (e.g., VOC analysis on only two groups), statistical concerns, and minor editorial issues that should be addressed. The novelty lies in the commercial-scale application and focus on flavor VOCs, but the findings on growth and carcass traits align with prior studies, reducing groundbreaking impact.
- Selective Analysis of VOCs and Flavor Profiles: The VOC analysis (Section 3.4–3.5) is limited to only the Lys100 and Lys145 groups, despite the study including four treatment groups. This is justified briefly by noting better carcass traits and amino acid profiles in Lys145, but it feels post-hoc and limits the ability to assess dose-response effects on flavor (e.g., linear/quadratic trends as done for other traits). Why not analyze all groups? If due to cost or resources, this should be explicitly stated and justified. Additionally, expanding to all groups could reveal if intermediate levels (e.g., Lys130) offer similar benefits, which would be practically useful for industry recommendations. Consider adding this or providing a stronger rationale.
2. Sample Size and Power: For growth performance, n=4 pens/group is adequate for pen-level metrics (e.g., ADG, ADFI). However, for carcass/meat quality (n=8 pigs/group) and VOCs (n=7–8), the sample size is small relative to the variability in commercial settings. Power calculations are absent—did the study have sufficient power to detect differences in VOCs or amino acids? For instance, several trends (e.g., backfat thickness, P=0.067–0.096) might become significant with larger n. Provide power analysis or discuss limitations in detecting subtle effects.
3. Ethical and Experimental Timeline: The Institutional Animal Care and Use Committee approval is dated 2011 (SKLAB-2011-04-03), which seems unusually old for a 2025 publication. Clarify if this is a typo or if the study was conducted much earlier—animal welfare standards have evolved (e.g., updated guidelines from NRC or EU directives). Also, confirm the experiment's duration (31 days) aligns with "finishing" phase definitions; pigs started at ~104 kg and ended at ~136 kg, which is reasonable but short—discuss if longer exposure might alter results. 4. Mechanistic Explanations in Discussion: The discussion speculates on mechanisms (e.g., lysine activating mTORC1, AMPK for fat deposition; lipid oxidation for VOCs) but lacks direct evidence from this study. For example, the upregulation of aldehydes like (E)-2-Nonenal and (E)-2-Octenal is attributed to potential accelerated lipid oxidation (Page 12), but no measures of oxidation (e.g., TBARS, fatty acid profiles) are provided. Suggest adding such analyses or toning down claims. Similarly, the link between increased methionine/glycine and flavor is plausible but not empirically tested here—consider citing metabolomics studies or proposing future work. 5. Novelty and Broader Implications: The study builds on prior work (e.g., refs [10–12,18]) showing lysine effects on carcass and IMF, but the flavor aspect under commercial conditions is novel. However, compare more explicitly to recent studies (post-2020) on amino acid modulation of pork volatiles (e.g., arginine or low-protein diets). Discuss practical implications: a 45% lysine increase raises costs—estimate economic trade-offs for flavor benefits. Also, address breed specificity (Duroc×Landrace×Yorkshire) vs. local breeds mentioned in intro (e.g., Laiwu, Ningxiang). 6. Please also mention about the health benefits of Lysine in human diet in introduction and discussion since the journal is Foods not Animal foods.
Some improvements are needed.
Author Response
- Selective Analysis of VOCs and Flavor Profiles: The VOC analysis (Section 3.4–3.5) is limited to only the Lys100 and Lys145 groups, despite the study including four treatment groups. This is justified briefly by noting better carcass traits and amino acid profiles in Lys145, but it feels post-hoc and limits the ability to assess dose-response effects on flavor (e.g., linear/quadratic trends as done for other traits). Why not analyze all groups? If due to cost or resources, this should be explicitly stated and justified. Additionally, expanding to all groups could reveal if intermediate levels (e.g., Lys130) offer similar benefits, which would be practically useful for industry recommendations. Consider adding this or providing a stronger rationale.
Response: Thanks for the reviewer’s suggestion. Due to the high cost and time-consuming nature of GC×GC-TOFMS analysis, we focused on the two extreme groups to maximize the contrast in flavor compound profiles. This approach was justified by the significant improvements in carcass traits and key amino acids (methionine and glycine) observed in the Lys145 group. We acknowledge that including all groups would provide a more complete dose-response relationship, and this limitation has been stated in the revised manuscript on Line 412-414 However, the observed linear trends in loin eye area and methionine content suggest that the benefits are dose-dependent, with the greatest improvement occurring at the highest lysine level. We agree that future studies should include VOC analysis across all treatment levels to better inform industry recommendations.
- Sample Size and Power: For growth performance, n=4 pens/group is adequate for pen-level metrics (e.g., ADG, ADFI). However, for carcass/meat quality (n=8 pigs/group) and VOCs (n=7–8), the sample size is small relative to the variability in commercial settings. Power calculations are absent—did the study have sufficient power to detect differences in VOCs or amino acids? For instance, several trends (e.g., backfat thickness, P=0.067–0.096) might become significant with larger n. Provide power analysis or discuss limitations in detecting subtle effects.
Response: Thanks for the reviewer’s suggestion. According to previous studies, our sample size is within the range commonly reported in high-impact meat science publications and is considered sufficient [1, 2]. Furthermore, as suggested by the reviewer, we conducted a post-hoc power analysis for key significant outcomes to quantify the strength of our findings, including lion eye area and the percentage of aldehydes (Table 1). This analysis was performed using G*Power 3.1 software, and the results showed that our study had sufficient power (>80%) to detect the significant effects reported for these primary outcomes. However, it should be noted that some traits, such as backfat thickness, showed strong decreasing trends with increasing lysine levels that did not reach strict statistical significance. This may be due to the inherent variability in commercial populations and the sample size, which may limited our ability to detect more subtle effects on all measured parameters. We have discussed this limitation on Line 357-361 in the revised manuscript.
Table 1. Results of the post-hoc power analysis conducted for the main significant findings.
Parameters |
Statistical test |
Comparison |
Effect size |
α-level |
Sample size |
Achieved power |
Lion eye area |
AONVA |
Four groups |
0.65 |
0.05 |
n=8 |
0.93 |
% of aldehydes |
T test |
Lys100 vs Lys145 |
1.45 |
0.05 |
n=8 |
0.89 |
References:
[1] Pan, J.; Han, S.; Wu, Q.; Chen, J.; Huang, R.; Guo, Q. Nutritional modulation of pork quality: SID lysine levels alter meat quality, and flavor-associated metabolites in growing-finishing pigs. Food Chem 2025, 493, 145620.
[2] Yu, H.; Chen, X.; He, J.; Zheng, P.; Luo, Y.; Yu, B.; Chen, D.; Huang, Z. Effects of dietary grape seed proanthocyanidin extract supplementation on muscle water-holding capacity of finishing pigs. Meat Sci 2025, 227, 109865.
- Ethical and Experimental Timeline: The Institutional Animal Care and Use Committee approval is dated 2011 (SKLAB-2011-04-03), which seems unusually old for a 2025 publication. Clarify if this is a typo or if the study was conducted much earlier—animal welfare standards have evolved (e.g., updated guidelines from NRC or EU directives). Also, confirm the experiment's duration (31 days) aligns with "finishing" phase definitions; pigs started at ~104 kg and ended at ~136 kg, which is reasonable but short—discuss if longer exposure might alter results.
Response: Thanks for the reviewer’s suggestion. The Institutional Animal Care and Use Committee approval number is AW01202313-1, and we have corrected it on Line 93in the revised manuscript. In this study, the 31-day experimental period is consistent with the finishing phase for pigs weighing 100–135 kg, as recommended by NRC (2012) and standard industry practices. Furthermore, the 31-day period is sufficient to observe changes in growth, carcass traits, and meat quality, as demonstrated in similar studies [1,2].
References:
[1] Atoo, A.F.; Levesque, C.L.; Thaler, R.; Underwood, K.; Beyer, E.; Perez-Palencia.; J.Y. Effect of feeding high oleic soybean oil to finishing pigs on growth performance, carcass characteristics, and meat quality. J Anim Sci 2025, 103, skae393.
[2] Lebret, B.; Batonon-Alavo. D.I.; Perruchot, M.H.; Mercier, Y.; Gondret, F. Improving pork quality traits by a short-term dietary hydroxy methionine supplementation at levels above growth requirements in finisher pigs. Meat Sci 2018, 145, 230-237.
- Mechanistic Explanations in Discussion: The discussion speculates on mechanisms (e.g., lysine activating mTORC1, AMPK for fat deposition; lipid oxidation for VOCs) but lacks direct evidence from this study. For example, the upregulation of aldehydes like (E)-2-Nonenal and (E)-2-Octenal is attributed to potential accelerated lipid oxidation (Page 12), but no measures of oxidation (e.g., TBARS, fatty acid profiles) are provided. Suggest adding such analyses or toning down claims. Similarly, the link between increased methionine/glycine and flavor is plausible but not empirically tested here—consider citing metabolomics studies or proposing future work.
Response: According to the reviewer’s suggestion, we have claimed that the increase in 1-Octen-3-one, (E)-2-Nonenal and (E)-2-Octenal may be related to enhanced lipid oxidation, and oxidation markers should be measured in future research. Furthermore, the contribution of increased methionine/glycine to flavor should be investigated. Please see details on Line 384-385 and 410-412 in the revised manuscript.
- Novelty and Broader Implications: The study builds on prior work (e.g., refs [10–12,18]) showing lysine effects on carcass and IMF, but the flavor aspect under commercial conditions is novel. However, compare more explicitly to recent studies (post-2020) on amino acid modulation of pork volatiles (e.g., arginine or low-protein diets). Discuss practical implications: a 45% lysine increase raises costs—estimate economic trade-offs for flavor benefits. Also, address breed specificity (Duroc×Landrace×Yorkshire) vs. local breeds mentioned in intro (e.g., Laiwu, Ningxiang).
Response: Thanks for the reviewer’s suggestion. Arginine exhibits synergistic effects with other amino acids. Supplementing with 1.0% L-arginine and 1.0% glutamic acid significantly increased IMF accumulation and fatty acid content, generating multiple VOCs derived from fatty acid oxidation. This enhances meat tenderness, juiciness, and overall eating quality [1]. Dietary low protein (14.0%) can also significantly alter the volatile compound profile of pork and increased abundance of aldehydes, ketones, and alcohols [2]. The novelty of this study lies in its first demonstration under commercial conditions that lysine optimizes flavor by regulating flavor precursors such as methionine and glycine, as well as key volatile organic compounds like aldehydes with ROAV≥1. We have added the above discussion on Line 61-65 in the revised manuscript. Furthermore, in this study, a 45% increase in lysine may raise feed costs, so an economic analysis is needed to balance flavor improvement with production costs. Our results are based on Duroc × Landrace × Yorkshire pigs, which are widely used in commercial production. However, responses may differ in local breeds due to genetic variations in metabolism and fat deposition. We have added the above discussion on Line 414-419 in the revised manuscript.
Reference:
[1] Guo, Q.; Kong, X.; Hu, C.; Zhou, B.; Wang, C.; Shen, Q. W. Fatty acid content, flavor compounds, and sensory quality of pork loin as affected by dietary supplementation with l-arginine and glutamic acid. J Food Sci 2019, 84, 3445-3453.
[2] Wang, D.; Hou, K.; Kong, M.; Zhang, W.; Li, W.; Geng, Y.; Ma, C.; Chen, G., Low-protein diet supplemented with amino acids can regulate the growth performance, meat quality, and flavor of the Bamei pigs. Foods 2025, 14.
- Please also mention about the health benefits of Lysine in human diet in introduction and discussion since the journal is Foods not Animal foods.
Response: Thanks for the reviewer’s suggestion. From a human nutritional perspective, lysine is an essential amino acid that must be obtained from food. It promotes protein synthesis, regulates fatty acid metabolism, aids calcium absorption, boosts immunity, and alleviates anxiety [1-3]. Please see details on Line 66-69 in the revised manuscript.
References:
[1] Gunarathne, R.; Guan, X.; Feng, T.; Zhao, Y.; Lu, J. L-lysine dietary supplementation for childhood and adolescent growth: Promises and precautions. J Adv Res 2025, 70, 571-586.
[2] Aggarwal, R.; Bains, K. Protein, lysine and vitamin D: Critical role in muscle and bone health. Crit Rev Food Sci Nutr 2022, 62, 2548-2559.
[3] Smriga, M.; Ghosh, S.; Mouneimne, Y.; Pellett, P. L.; Scrimshaw, N. S. Lysine fortification reduces anxiety and lessens stress in family members in economically weak communities in Northwest Syria. Proc Natl Acad Sci U S A 2004, 101, 8285-8288.
Reviewer 2 Report
Comments and Suggestions for Authors
The title should be interconnected with the study's objective and present what was evaluated in this research. I suggest rewriting
Abstract:
I recommend formulating the objective as a direct sentence, preferably linked to the research hypothesis;
What is the experimental design? What are the treatments? How many replicates?
What is the experimental period?
Note any inconsistencies between the results presented in the abstract and within the text. Present probability values.
Keywords: It is incorrect to repeat words contained in the title. Try to include variables that were highlighted in the research in this section, as the title can also be used to search for similar papers, so greater search possibilities with different keywords allow for a better search.
Introduction
Information related to pork production/consumption worldwide or in the country is recommended for your introduction.
I believe that a presentation related to how lysine levels above the recommended level can alter protein or lipid metabolism would help justify the answers obtained in this study. All abbreviations must be defined before use. Check this throughout the paper.
What gaps exist in the various existing studies that will be investigated in this paper? What innovation justifies the research being publishable? Why investigate lysine beyond the known requirements? What is already known (or unknown) about its impact on VOCs and lipid profile?
Lines 70-72 – Unnecessary. The authors present the expected results and impacts, and this is incorrect. This information should be presented in research projects, not at this stage.
What is the study hypothesis?
The objective should be clearly stated.
Materials and methods
The chemical composition of the experimental diets should be included.
How were the zootechnical performance variables calculated? Include the equations.
How was slaughter performed? Following what recommendations? Describe.
Check the standards of this journal and adjust the presentation of the equations.
Clearly indicate the number of animals per treatment in the methodology.
In color analysis, what is the angle of observation? What methodology was used? I suggest calculating saturation and hue.
For all methodologies presented, include the citation and reference.
Item 2.4: The methodologies need to be better detailed and the protocol numbers presented.
Methodologies should be presented for all variables presented in the results, as well as all equations used to estimate the variables. Check.
The statistical analysis needs to be clearer and all data analysis steps described. The statistical model should be included.
Author Response
Reviewer #2: The title should be interconnected with the study's objective and present what was evaluated in this research. I suggest rewriting.
Response: Thanks for the reviewer’s suggestion. We have revised the title to “Dietary lysine supplementation above requirement improves carcass traits and enhances pork flavor profiles in finishing pigs under commercial conditions” in the revised manuscript.
Abstract:
(1) I recommend formulating the objective as a direct sentence, preferably linked to the research hypothesis; What is the experimental design? What are the treatments? How many replicates? What is the experimental period?
Response: According to the reviewer’s suggestion, we have added the above information on Line 11-19 in the revised manuscript.
(2) Note any inconsistencies between the results presented in the abstract and within the text. Present probability values.
Response: Thanks for the reviewer’s reminder. We have included the relevant additions in the abstract of the revised manuscript.
(3) Keywords: It is incorrect to repeat words contained in the title. Try to include variables that were highlighted in the research in this section, as the title can also be used to search for similar papers, so greater search possibilities with different keywords allow for a better search.
Response: Thanks for the reviewer’s suggestion. We have revised the keywords to “Carcass traits; Dietary lysine levels; Free amino acids; Pork quality; Volatile flavor compounds” on Line 33-34 in the revised manuscript.
Introduction
(1) Information related to pork production/consumption worldwide or in the country is recommended for your introduction.
Response: Thanks for the reviewer’s suggestion. In 2024, the global production of pork reached 116.02 billion tonnes. China, the European Union and the United States accounted for 48.91%, 18.32% and 10.93% (United States Department of Agriculture, USDA). We have added it on Line 37-39 in the revised manuscript.
(2) I believe that a presentation related to how lysine levels above the recommended level can alter protein or lipid metabolism would help justify the answers obtained in this study. All abbreviations must be defined before use. Check this throughout the paper.
Response: Thanks for the reviewer’s suggestion. Lysine is a core driver of protein synthesis, promoting satellite cell differentiation by activating the mTORC1 signaling pathway and increasing skeletal muscle mass [1]. In contrast, the expression levels of AMPKα1, SIRT1 and PGC-1α were significantly reduced under lysine restriction, leading to inhibited lipolysis [2]. Furthermore, we have also confirmed that all abbreviations are defined when they are first used throughout the manuscript.
References:
[1] Jin, C.; Zhang, Z.; Song, Z.; Gao, C.; Yan, H.; Wang, X. mTORC1-mediated satellite cell differentiation is required for lysine-induced skeletal muscle growth. J Agric Food Chem 2020, 68, 4884-4892.
[2] Wang, J.; Li, H.; Zhu, H.; Xia, S.; Zhang, F.; Zhang, H.; Liu, C.; Zheng, W.; Yao, W. Impacts of dietary standardized ileal digestible lysine to net energy ratio on lipid metabolism in finishing pigs fed high-wheat diets. Animals (Basel) 2024, 14, 1824.
(3) What gaps exist in the various existing studies that will be investigated in this paper? What innovation justifies the research being publishable? Why investigate lysine beyond the known requirements? What is already known (or unknown) about its impact on VOCs and lipid profile?
Response: Thanks for the reviewer’s suggestion.
â‘ Studies regarding the effects of dietary lysine levels on pork flavor characteristics are currently limited. Furthermore, most studies have used small numbers of pigs, and the impact of dietary lysine levels (DLLs) on pork quality and flavor under large-scale commercial farming conditions requires systematic investigation. Please see details on Line 357-361 in the revised manuscript.
â‘¡ The key innovation is demonstrating that lysine supplementation up to 145% of requirement under commercial conditions simultaneously improves both carcass leanness and sensory flavor. This is demonstrated using advanced GC×GC-TOFMS to identify the key flavor compounds responsible, providing a practical, evidence-based solution for producing premium pork.
â‘¢ Lysine is the first limiting amino acid in pig diets. Decreasing DLLs below the required amount would compromise growth performance. This would reduce the likelihood of its application in pig farms. Therefore, we investigated lysine beyond the known requirements.
â‘£ Lysine has a sweet taste, and the carnosine formed from lysine could mask bitterness. Additionally, lysine promotes carnitine synthesis, enhances fatty acid β-oxidation and improves ATP production efficiency, thereby increasing inosine monophosphate (IMP) accumulation, which is a key source of umami flavor in meat. Therefore, DLLs may influence the development of pork flavor. However, studies regarding the effects of DLLs on pork flavor characteristics are currently limited. Please see details on Line 76-84 in the revised manuscript.
(4) Lines 70-72 – Unnecessary. The authors present the expected results and impacts, and this is incorrect. This information should be presented in research projects, not at this stage.
Response: According to the reviewer’s suggestion, we have deleted this sentence in the revised manuscript.
(5) What is the study hypothesis?
Response: Thanks for the reviewer’s suggestion. We hypothesized that increasing dietary lysine levels would alter pork flavor without negatively affecting growth performance, carcass traits or meat quality under large-scale farming conditions. We have added them on Line 87-89 in the revised manuscript.
(6) The objective should be clearly stated.
Response: The objective of this study was to investigate the effects of dietary lysine levels on carcass traits, meat quality and VOCs composition in pork while meeting the lysine requirements, using 450 finishing pigs. We have clearly stated it on Line 85-87 in the revised manuscript.
Materials and methods
(1) The chemical composition of the experimental diets should be included.
Response: The chemical composition of the experimental diets, including crude protein content and the levels of SID amino acids, calcium and available phosphorus, has been included in Table 1. Please see details in the revised manuscript.
(2) How were the zootechnical performance variables calculated? Include the equations.
Response: The calculation formulas are follows: ADG (kg/d) = [Final BW (kg)-Initial BW (kg)]/31(d), Average daily feed intake (kg/d) (ADFI)=Total feed intake (kg)/31(d), Ratio of feed to gain= ADFI (kg/d)/ ADG (kg/d). We have added them on Line 110-112 in the revised manuscript.
(3) How was slaughter performed? Following what recommendations? Describe.
Response: Thanks for the reviewer’s suggestion. The slaughter procedure was conducted strictly in accordance with World Organization for Animal Health (OIE) guidelines on animal welfare and standard commercial practices, in order to minimize stress and ensure sample integrity. Following a 12-hour fasting period with free access to water, pigs were transported to a nearby licensed abattoir. They were rested for at least four hours and then slaughtered using electrical stunning followed by exsanguination. Please see details on Line 115-119 in the revised manuscript.
(4) Check the standards of this journal and adjust the presentation of the equations.
Response: Thanks for the reviewer’s reminder. We have made corresponding modifications in the revised manuscript. We have also carefully checked the entire manuscript for similar issues.
(5) Clearly indicate the number of animals per treatment in the methodology.
Response: Thanks for the reviewer’s reminder. A total of approximately 110 pigs were involved in each group. We have added it on Line 98-99 in the revised manuscript.
(6) In color analysis, what is the angle of observation? What methodology was used? I suggest calculating saturation and hue.
Response: Thanks for the reviewer’s professional suggestion. After a 30-min blooming process, the meat color was evaluated using triple-stimulus colorimeter (CR-410, Minolta, Japan) with an 50 mm measuring aperture illuminated by a standardized xenon lamp. The instrument was set up with a D65 light source and a 2° standard observer to determine the lightness (L*), redness (a*), and yellowness (b*) values at 24 h post slaughter according to the CIE Lab system. Furthermore, according to the reviewer’s suggestion, we have supplemented the calculation of C* and h° in the Materials and Methods and Table 4. The calculation formula is as follows: , . Please see details on Line 156-158 and table 4 in the revised manuscript.
(7) For all methodologies presented, include the citation and reference.
Response: Thanks for the reviewer’s suggestion. We have supplemented the relevant citations and references in the revised manuscript. Please see details on materials and methods in the revised manuscript.
(8) Item 2.4: The methodologies need to be better detailed and the protocol numbers presented.
Response: Approximately 10 g of LT muscle samples (n=8) were accurately weighed. Moisture content was determined by vacuum freeze-drying the samples at -50°C under 0.1 mbar for 48 h (Christ Alpha 1-4 LDplus, Martin Christ Gefriertrocknungsanlagen GmbH, Osterode am Harz, Germany), followed by reweighing. Moisture content was calculated as follows: [(Initial weight – Freeze-dried weight)/Initial weight]×100%. For crude protein (CP) determination, freeze-dried muscle samples were ground into a homogeneous powder, and then analyzed via the Kjeldahl method (AOAC, 2007; method 984.13) using a Kjeltec 8400 Auto Analyzer (FOSS Analytical A/S, Hillerød, Denmark). CP content was calculated by multiplying the nitrogen content by a conversion factor of 6.25. IMF was extracted using a Budwi Extraction System B-11 (Budwi, Lausanne, Switzerland) based on the Soxhlet extraction principle (AOAC, 2007; method 991.36). Ground freeze-dried samples were wrapped in filter paper thimbles, extracted with petroleum ether (boiling range 30–60°C) for 6 h, then dried at 105°C for 1 h and cooled in a desiccator before weighing. IMF content was expressed as a percentage relative to the initial fresh weight of the muscle. We have added these details to Section 2.4 according to the reviewer’s suggestion.
(9) Methodologies should be presented for all variables presented in the results, as well as all equations used to estimate the variables. Check.
Response: Thanks for the reviewer’s reminder. In the revised manuscript, we have included the formulae for calculating average daily weight gain, average daily feed intake and feed conversion ratio, loin eye area, chroma, hue angle, moisture content and relative odor activity value. Please see details on Line110-112, 143-144, 158 and 228-230 in the revised manuscript.
(10) The statistical analysis needs to be clearer and all data analysis steps described. The statistical model should be included.
Response: Thanks for the reviewer’s reminder. We have clearly described the statistical analysis and included the statistical model in the revised manuscript. Please see details on Line 222-228 and 243-244 in the revised manuscript.
Reviewer 3 Report
Comments and Suggestions for Authors​ 1. According to the authors, this article explores the effects of dietary lysine levels (DLLs) on growth performance, carcass traits, meat quality, and flavor characteristics in finishing pigs under large-scale commercial farming conditions.
What is the aim of the study? When comparing the objective in the abstract and in the introduction, it is not the same.
Therefore, while meeting the lysine requirements, this study investigated the effects of DLLs on carcass traits, meat quality, and VOCs composition in pork, using 450 finishing pigs. The results of this study will provide a theoretical basis for determining dietary lysine parameters that balance growth performance and meat quality in finishing pigs.
Line 13: "The results showed" - that we had to evaluate the results from the data analysis before.
2. Does the original topic have a practical application? Resulting from the relationship between the pig diet and digestible lysine and pig meat quality.
For consideration, will you compare and discuss the potential differences in results obtained from a small number of pigs and large-scale commercial farming? They could be, but we don't know, so next time, you propose to study 1000 pigs. Based on the obtained results, you can prove potential differences in the scale of samples.
3. The authors in the discussion paragraph present the state of knowledge on the level of lysine in the diet of other animals than pigs. What is the achievement of the present paper in this area? You should add a paragraph concerning the state of knowledge on this subject, what is confirmed, and your added value. (line 312…).
An additional table should show the pros and cons of increasing the lysine level in the pork diet.
4. The manuscript is correctly prepared. However, the discussion paragraph should be redesigned to give what the achievements of research are, what they add to present knowledge in the application of lysine in the diet of different animals, mainly pork, and what practical information is for pig farming. Moreover, in the conclusion paragraph, we are informed that 45% increase in lysine levels has some beneficial effects on flavour in the loin eye area and reduced backfat thickness. Still, it is hard to estimate the practical importance of its influence.
5. The conclusions are consistent with the evidence and arguments. The statement: This study provides valuable insights for optimizing nutritional regulation to improve pig production under commercial farming conditions, is the authors' opinion. Have you discussed the results with farmers about the potential application of changes in pig diet to get any profits (economic, more satisfactory client consuming such meat, or considering the well-being of animals?
6. The references look appropriate - please check all.
Sun, X.; Yu, Y.; Saleh, A. S. M.; Yang, X.; Ma, J.; Zhang, D.; Li, W.; Wang, Z. Comprehensive characterisation of taste and aroma 473 profiles of Daokou red-cooked chicken by GC-IMS and GC-MS combined with chemometrics. International J. Food Sci Technol 2023, 58, 8, 4288-4300.
7. The tables and figures are clear and well-presented.
Author Response
Reviewer #3:
- According to the authors, this article explores the effects of dietary lysine levels (DLLs) on growth performance, carcass traits, meat quality, and flavor characteristics in finishing pigs under large-scale commercial farming conditions.
What is the aim of the study? When comparing the objective in the abstract and in the introduction, it is not the same.
Therefore, while meeting the lysine requirements, this study investigated the effects of DLLs on carcass traits, meat quality, and VOCs composition in pork, using 450 finishing pigs. The results of this study will provide a theoretical basis for determining dietary lysine parameters that balance growth performance and meat quality in finishing pigs.
Line 13: "The results showed" - that we had to evaluate the results from the data analysis before.
Response: Thanks for the reviewer’s suggestion. The aim of this study was to explore the effects of DLLs on growth performance, carcass traits, meat quality, and flavor characteristics in finishing pigs under large-scale commercial farming conditions. We have clearly stated it on Line 11-13 in the abstract and on Line 85-87 in Introduction. Furthermore, we have precisely described the experimental design before “The results showed”. Please see details on Line 13-19in the revised manuscript.
- Does the original topic have a practical application? Resulting from the relationship between the pig diet and digestible lysine and pig meat quality.
For consideration, will you compare and discuss the potential differences in results obtained from a small number of pigs and large-scale commercial farming? They could be, but we don't know, so next time, you propose to study 1000 pigs. Based on the obtained results, you can prove potential differences in the scale of samples.
Response: Thank you for your valuable feedback. Our results demonstrate that strategic nutritional modulation of digestible lysine is a viable tool to improve two key economic factors: carcass leanness and meat flavor. By increasing the dietary lysine level (e.g., 145% of the required amount), producers can produce a premium product with an enhanced eating quality without compromising growth performance. This aligns with the growing market demands for high-quality, flavorful pork. The economic trade-off between the increased cost of supplemental amino acids and the potential premium for high-quality meat is a tangible calculation for the industry.
Furthermore, we greatly appreciate the reviewer's insight into the differences between controlled experiments and large-scale farming. While our study was conducted on a commercial farm, it still employed a controlled experimental design with replicated pens. However, we cannot prove with absolute certainty that the same magnitude of effect would be observed on every 1,000-pig farm. Variability in management, health status, genetics and environmental conditions across commercial units introduces inherent uncertainty. In our study, however, the pigs were housed in standard production facilities, common management practices were employed, and the pigs were fed diets formulated with commercially available ingredients. This significantly enhances the external validity and scalability of our findings compared to studies conducted in small-scale research stations. The effects we observed were statistically significant and consistent with the known role of lysine in protein synthesis. In summary, our research has clear practical applications for formulating diets aimed at producing premium pork. While we are confident in the validity of our results, we agree that large-scale validation is key to achieving universal industry adoption.
- The authors in the discussion paragraph present the state of knowledge on the level of lysine in the diet of other animals than pigs. What is the achievement of the present paper in this area? You should add a paragraph concerning the state of knowledge on this subject, what is confirmed, and your added value. (line 312…).
An additional table should show the pros and cons of increasing the lysine level in the pork diet.
Response: Thanks for the reviewer’s suggestion. The key achievement of this paper in this area is extending the lysine metabolic pathway associations identified in other research to flavor modulation in commercially farmed pigs. It demonstrates that unlike other species, pigs exhibit reduced backfat (without altering intramuscular fat) and improved flavor via regulated flavor precursors (glycine) and volatile organic compounds (VOCs) when fed supra-recommended lysine. This fills the gap between basic lysine research in other animals and applied flavor optimization in swine production, providing pig-specific empirical evidence for cross-species amino acid nutrition studies. We have added it on Line 355-357 in the revised manuscript.
- The manuscript is correctly prepared. However, the discussion paragraph should be redesigned to give what the achievements of research are, what they add to present knowledge in the application of lysine in the diet of different animals, mainly pork, and what practical information is for pig farming.
Moreover, in the conclusion paragraph, we are informed that 45% increase in lysine levels has some beneficial effects on flavor, in the loin eye area and reduced backfat thickness. Still, it is hard to estimate the practical importance of its influence.
Response: We have redesigned the discussion paragraph according to the reviewer’s suggestion. Furthermore, a 45% increase in lysine increases lion eye area, reduces backfat thickness, and enhances the sweetness, fruity, fatty and waxy flavor characteristics of pork without compromising growth performance. However, it should be noted that a 45% increase in lysine may raise feed costs, so an economic analysis is needed to balance flavor improvement with production costs. Its application requires both large-scale trials to validate economic benefits. We have discussed it on Line 412-419 in the revised manuscript.
- The conclusions are consistent with the evidence and arguments. The statement: This study provides valuable insights for optimizing nutritional regulation to improve pig production under commercial farming conditions, is the authors' opinion. Have you discussed the results with farmers about the potential application of changes in pig diet to get any profits (economic, more satisfactory client consuming such meat, or considering the well-being of animals?
Response: Thanks for the reviewer’s suggestion. Our findings indicate that increasing SID lysine levels to 45% of the NRC (2012) recommended value increases lion eye area, reduces backfat thickness, and enhances the sweetness, fruity, fatty and waxy flavor characteristics of pork without compromising growth performance. This discovery has direct implications for producing premium-quality pork. However, it should be noted that a 45% increase in lysine may raise feed costs, so an economic analysis is needed to balance flavor improvement with production costs. We have addressed this issue on Line 412-417 in the revised manuscript. In the future, we intend to conduct interviews with farmers to validate the economic benefits of the lysine optimisation scheme, including savings on feed costs and premium pricing for high-quality pork, as well as its practical impact on consumer satisfaction and animal welfare.
- The references look appropriate - please check all.
Sun, X.; Yu, Y.; Saleh, A. S. M.; Yang, X.; Ma, J.; Zhang, D.; Li, W.; Wang, Z. Comprehensive characterisation of taste and aroma 473 profiles of Daokou red-cooked chicken by GC-IMS and GC-MS combined with chemometrics. International J. Food Sci Technol 2023, 58, 8, 4288-4300.
Response: Thanks for the reviewer’s reminder. We have corrected this reference on Line 543-545 and carefully reviewed the entire manuscript once again.
- The tables and figures are clear and well-presented.
Response: Thank you for the appreciation of our work by the reviewer.
Round 2
Reviewer 2 Report
Comments and Suggestions for Authors
The authors have made all the requested adjustments to improve the paper. I recommend that the paper be accepted for publication.
Author Response
Thank you for the positive feedback and for accepting our manuscript for publication.
Reviewer 3 Report
Comments and Suggestions for Authors
The author's explanations are satisfactory. However, there is still a lack of a figure or table with pros and cons concerning the current knowledge about the application of lysine in pigs' diets.
Author Response
The author's explanations are satisfactory. However, there is still a lack of a figure or table with pros and cons concerning the current knowledge about the application of lysine in pigs' diets.
Response: We sincerely thank the reviewer for this positive and constructive suggestion. We agree that a concise summary table will greatly enhance the clarity and impact of our manuscript. As suggested, we have summarized the pros and cons of increasing the lysine level in the pig diet as following. We have added it as Table S3 in the revised manuscript.
